# Emergency mortality of non-trauma patients was predicted by qSOFA score

**Yufang Li** [ID]*ᵒ, **Yanxia Guo**ᵒ, **Du Chen***

Department of Critical Care Medicine, The First Affiliated Hospital of Soochow University, Suzhou City, Jiangsu Province, China

ᵒ These authors contributed equally to this work.
* sdfyycd@suda.edu.cn

**Data Availability Statement:** All relevant data are within the manuscript and its Supporting Information files.

**Funding:** The author(s) received no specific funding for this work.

## Abstract

### Objective

This study was aimed to evaluate the performance of quick sequential organ failure assessment (qSOFA) in predicting the emergency department (ED) mortality of non-trauma patients and to expand the application scope of qSOFA score.

### Methods

A single, retrospective review of non-trauma patients was conducted in ED between November 1, 2016 and November 1, 2019. The qSOFA score was obtained from vital signs and Glasgow Coma Scale (GCS) score. The outcome was ED mortality. Multivariable logistic regression analysis was performed to explore the association between the qSOFA score and ED mortality. The area under the receiver operating characteristic (AUROC) curve, the best cutoff value, sensitivity and specificity were performed to ascertain the predictive value of the qSOFA score.

### Results

228(1.96%) of the 11621 patients were died. The qSOFA score was statistically higher in the non-survival group (P<0.001). The qSOFA score 0 subgroup was used as reference baseline, after adjusting for gender and age, adjusted OR of 1, 2 and 3 subgroups were 4.77 (95%CI 3.40 to 6.70), 18.17 (95%CI 12.49 to 26.44) and 23.63 (95%CI 9.54 to 58.52). All these three subgroups show significantly higher ED mortality compared to qSOFA 0 subgroup (P<0.001). AUROC of qSOFA score was 0.76 (95% CI 0.73 to 0.79). The best cutoff value was 0, sensitivity was 77.63% (95%CI 71.7% to 82.9%), and specificity was 67.2% (95%CI 66.3% to 68.1%).

### Conclusion

The qSOFA score was associated with ED mortality in non-trauma patients and showed good prognostic performance. It can be used as a general tool to evaluate non-trauma patients in ED. This is just a retrospective cohort study, a prospective or a randomized study will be required.

**Competing interests:** The authors have declared that no competing interests exist.

## Introduction

The emergency department (ED) is a place with many patients, whose illness are diverse and complex, especially non-trauma patients. Emergency physicians and nurses need to focus on critically ill patients, so rapid and accurate identification of patients at high risk of death is the crux to reduce ED mortality and optimize health resources. A variety of clinical tools have been proven to assess outcomes in ED patients, such as modified early warning score (MEWS), national early warning score(NEWS) and Acute Physiology and Chronic Health Evaluation (APACHEⅡ) [1–3]. However, it is inconvenient to use them in a fast-paced ED, because of the cumbersome evaluation process or containing laboratory indicators.

In 2016, a novel tool was proposed to identify patients at high risk of death from sepsis, which was the Quick Sepsis-Related Organ Failure Assessment (qSOFA) [4]. Although practicality remains controversial, several recent studies suggested that qSOFA can predict mortality in noninfectious disease, such as pancreatitis, cirrhosis, Glyphosate poisoning and trauma [5–8]. Because most of the non-trauma patients in the ED died of respiratory failure, circulatory failure and other organ dysfunction, which were included in the definition of "life-threatening organ dysfunction" in sepsis [4,9].

A simple, universal tool that can be quickly calculated at the bedside without any biochemical markers or waiting time will be very useful for emergency physicians to assess patients. It was found that qSOFA fully meets these requirements. It is so easy to remember, because it comprises only three binary variables. Therefore, we used the qSOFA score to conduct a mortality study of non-trauma patients in ED.

## Materials and methods

### Design and setting

This retrospective cohort study was conducted at the First Affiliated Hospital of Soochow University, a general hospital in China. The data of this study were taken from the hospital electronic medical record system and all the data of patients were collected without identifiable personal information. The study was approved by the Ethics Committees of the First Affiliated Hospital of Soochow University (Ethical Research Batch NO.2020152). The Institutional Review Boards of the Ethics Committees waived the need for informed consent from our study participants due to the retrospective nature of the data. This study conforms to the principles outlined in the Declaration of Helsinki.

### Selection of participants

The study was conducted from November 1, 2016 to November 1, 2019. The non-trauma patients who were treated by the physician in the ED were selected using the hospital electronic medical record system. Patients who were under the age of 18 and had a prior terminal illness were excluded in the study. We also excluded dental, dermatological disease, traumatic disease and neurological diseases since they were not managed by the physicians in our ED.

### Data collection and processing

The following data were collected from the electronic medical records: age, gender, initial vital signs (systolic blood pressure, respiratory rate, pulse rate, temperature and oximetry), hours in the emergency room (HER) and Glasgow Coma Scale (GCS) score. Researchers can calculate the qSOFA score from these simultaneous (within 2 minutes) records. The qSOFA score was assessed and allocated one point for each of the three variables [4]: systolic BP $\leq$100 mm Hg, respiratory rate $\geq$22 breaths per minute, and altered mental status (GCS score $\leq$13).

## Outcome measures

The only outcome for this study was ED mortality. Death after hospitalization and discharge was not recorded. According to the qSOFA score, the patients were divided into four sub-groups to assess the association between outcomes and qSOFA.

## Data analysis

Continuous variables were tested for normality distribution using Shapiro-Wilk test. Continuous variables of non-normality were expressed as median (P25, P75) and compared using Mann-Whitney test. Categorical variables were expressed as frequencies and percentages and compared using Likelihood-ratio Chi squared test. Logistic regression models were performed to calculate the odds ratios (ORs) of variables for death. Receiver operating characteristic curve were performed to get the area under the curve (AUC). Statistics and plotting were completed by STATA 15.0 software. Two-tailed $P < 0.05$ was considered to be statistically significant.

## Results

### Comparison of survival group and non-survival group

A total of 11621 patients were involved in the study, of whom 7140(61.44%) were male. 11393 (98.04%) patients survived, and 228(1.96%) patients died. The mean age was 62 (47, 73) and 66 (54, 76) respectively. Patients in the survival group were younger than those who were in the non-survival group. And was statistically significant (P = 0.001). The HER were 11 (2, 24) and 7 (2, 22) in the survival and non-survival groups. However, the difference was not statistically significant (P = 0.325) (Table 1).

### ED Mortality according to qSOFA score

Among all the patients, 7708 patients had a qSOFA score of 0 (66.33%), 3241 patients had a qSOFA score of 1 (27.89%), 626 patients had a qSOFA score of 2 (5.39%), and 46 patients had a qSOFA score of 3 (0.39%). The qSOFA score was significantly higher in the non-survival group (P<0.001) (Table 1). According to the qSOFA score, the ED mortality of subgroup 0, 1,

**Table 1. Baseline characteristics.**

| Variables | Survival 11393(98.04%) | Non-survival 228(1.96%) | P value |
|---|---|---|---|
| Sex | | | 0.040 |
| Female | 4408(38.69%) | 73(32.02%) | |
| Male | 6985(61.31%) | 155(67.98%) | |
| Age (year) | 62(47, 73) | 66(54, 76) | 0.001 |
| MEWS | 2(1,4) | 6(3,10) | <0.001 |
| qSOFA | 0(0, 1) | 1(1, 2) | <0.001 |
| 0 | 7657(99.34%) | 51(0.66%) | |
| 1 | 3138(96.82%) | 103(3.18%) | |
| 2 | 558(89.14%) | 68(10.86%) | |
| 3 | 40(86.96) | 6(13.04%) | |
| HER (hour) | 11(2, 24) | 7(2, 22) | 0.325 |

qSOFA, quick sepsis-related organ failure assessment; MEWS, modified early warning score; HER, hours in the emergency room.

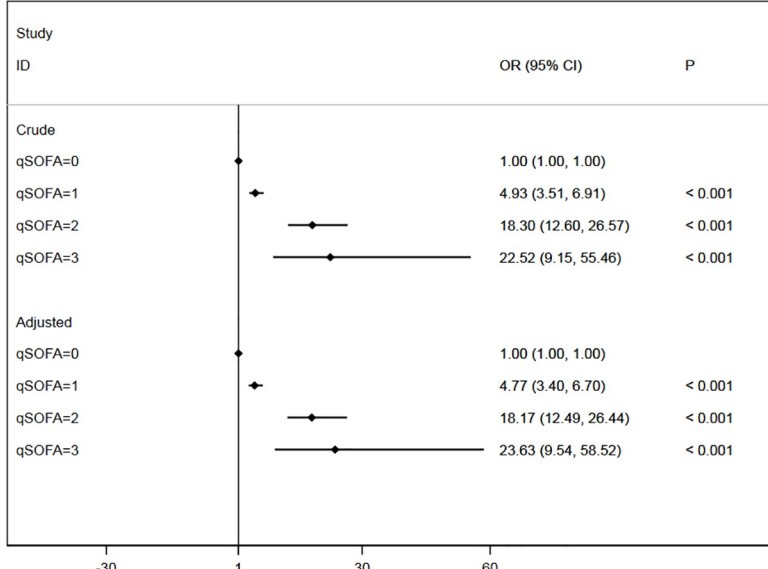

**Fig 1. Forest plot of crude ORs and adjusted ORs of qSOFA.** OR, odds ratio; Crude ORs were calculated by univariable logistic regression model. Adjusted ORs were calculated by multivariable logistic regression of qSOFA, sex, age.

2, and 3 were 0.6%, 3.28%, 12.06% and 15.38% respectively. Accompany with the increase of the qSOFA score, there is a trend of linear increase in ED mortality.

## Predictive performance

The patients were divided into four subgroups according to the qSOFA score. The qSOFA score 0 subgroup was used as reference baseline, the crude ORs of 1, 2 and 3 subgroups were 4.93 (95%CI 3.51 to 6.91), 18.30 (95%CI 12.60 to 26.57) and 22.52 (95%CI 9.15 to 55.46) respectively. After adjusting for gender and age, adjusted ORs of 1, 2 and 3 subgroups were 4.77 (95%CI 3.40 to 6.70), 18.17 (95%CI 12.49 to 26.44) and 23.63 (95%CI 9.54 to 58.52). All these three subgroups show significantly higher ED mortality compared to qSOFA 0 (P<0.001) (Fig 1). ED mortality was associated with the qSOFA score either before or after adjustment for other factors, and an increase in score predicted an increase in mortality. Area under the Receiver operating characteristic curve (AUROC) of qSOFA for predicting ED mortality in non-trauma patients was 0.76 (95% CI 0.73 to 0.79). The best cutoff value determined by Youden index was 0, the corresponding sensitivity was 77.63% (95%CI 71.7% to 82.9%), and specificity was 67.2% (95%CI 66.3% to 68.1%). Meanwhile, the AUROC for MEWS score was 0.79 (95% CI 0.76 to 0.83). The best cutoff value was 4, sensitivity was 64.0% (95%CI 57.4% to 70.3%), and specificity was 86.1% (95%CI 85.4% to 86.7%) (Fig 2). The difference was statistically significant (P = 0.003).

## Discussion

As soon as possible, the early identification and rapid response of acute coronary syndrome, sepsis and other critically ill patients with potential death risk can reduce the mortality [10]. This is one of the few studies to clarify the role of qSOFA score in predicting ED mortality in non-trauma patients.

It is shown that the qSOFA score in the ED can predict the mortality of non-trauma patients. The qSOFA score was associated with ED mortality (0 [0.66%], 1 [3.18%], 2 [10.86%],

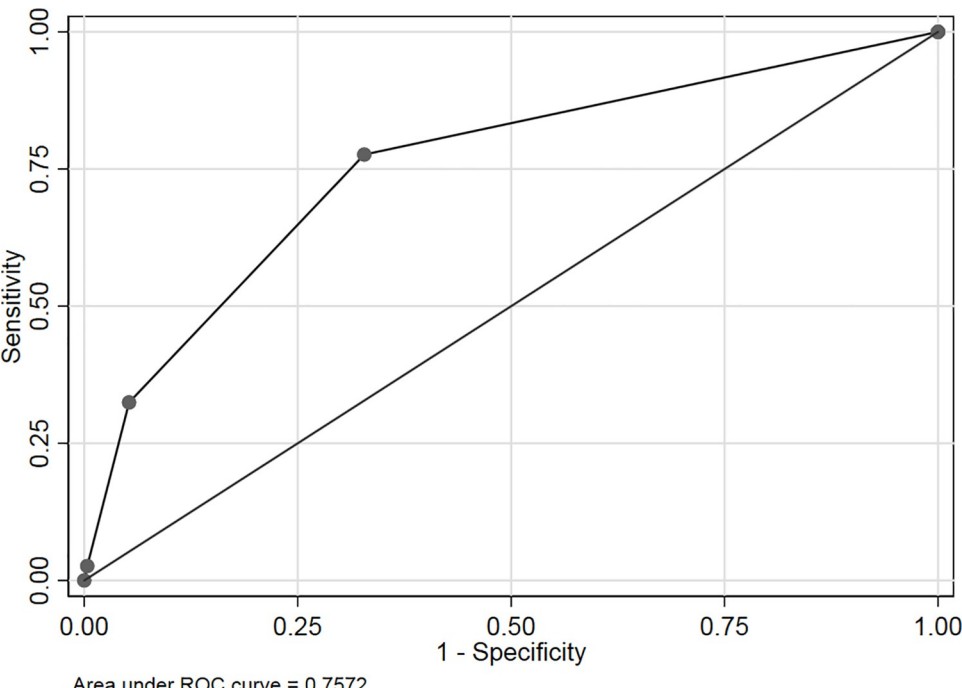

**Fig 2. Receiver operating characteristic curve of qSOFA and MEWS predicting death.**

and 3[13.04%]). As the qSOFA score increased from 0 to 3, the ED mortality significantly increased (P<0.001). Thus, the qSOFA score, which can be easily and rapidly calculated, can potentially be used as a general tool to predict death for emergency non-trauma patients.

The qSOFA score has been used as a new tool for sepsis and septic shock [4]. It was developed as a risk stratification tool for patients who are admitted outside the ICU, such as those in ED [11–13]. Seymour et al [11] studied 148,907 suspected infected patients, of whom 4% died. Among encounters with suspected infection outside of the ICU, the predictive validity (AUROC = 0.81; 95%CI, 0.80–0.82) for in-hospital mortality of qSOFA was statistically greater than SOFA and systemic inflammatory response syndrome (SIRS) criteria.

This report led to a bunch of studies in regard to the usefulness of the qSOFA score outside the ICU, some of which were on non-infectious diseases [6–8,12,13]. In patients with glyphosate surfactant herbicide poisoning [7], qSOFA was independently associated with in-hospital mortality. The AUC of qSOFA was as high as 0.841 (95% CI, 0.772–0.895). Singer AJ et al [12] further support its practicability, especially in undifferentiated ED patients. The study included 22,530 patients, who were divided into two subgroups: infected and uninfected. The predictive validity of death in patients with and without suspected infection were similarly high (0.75, 95% CI 0.71 to 0.78 and 0.70, 95% CI 0.65 to 0.74). They were consistent with our study (AUROC = 0.76, 95% CI 0.73 to 0.79). The difference was that our study did not distinguish between infection patients and non-infection ones.

A variety of clinical tools have been demonstrated to predict mortality of patients in ED. APACHEⅡis one of the most commonly used tools, whose accuracy had been confirmed in a large number of clinical studies [14,15]. The disadvantages of the APACHEⅡwere the complex calculation method and depends on laboratory testing [2]. That was not conducive to assess the condition quickly, so it was difficult to be widely used in ED [16].The MEWS and NEWS were also often used to predict the outcome of emergency patients. The MEWS was

miscalculated frequently, the probability as high as 18.2% [17]. The potential of NEWS implementation in clinical practice was proved complicated [18]. In addition, there were a number of potential false positives in patients with chronic hypoxemia when using NEWS as the tool [19]. With the application of computer technology, data can be automatically collected and analyzed, these problems can be well solved.

Our study shows that AUROC of qSOFA were lower than MEWS score, the difference was statistically significant. The MEWS score may be a better choice as a triage tool. The performance of the qSOFA in our study was similar to or even better than previous studies [14,20,21]. The advantages of qSOFA over other scores were that it contains only three binary variables and does not require complicated calculations. Therefore, the qSOFA score can be used as a tool to predict ED mortality in non-trauma patients.

This study still had several limitations. First, it was a retrospective study, which is subject to selection bias and errors of documentation and data entry. Secondly, the data were derived from a single institution and can't represent other Settings. Thirdly, the patients who died after hospitalization were not counted as deaths. It is not clear whether these patients are in some way different from those in this cohort, making our conclusions unreliable. Fourthly, we included only non-trauma patients managed by the physicians in ED, neurological diseases were excluded in the study. However, the qSOFA may overperform in population of neurological diseases, in which the ED death was more common. Future multicenter prospective studies need to be continued to complement this study.

In conclusion, the qSOFA score was associated with ED mortality in non-trauma patients, and an increase in score predicted an increase in ED mortality. The qSOFA score is a simple and valuable tool that can be used in ED to predict mortality. Further multicenter prospective studies of the qSOFA score are required before it can be widely used.

## Supporting information

**S1 Data.**
(XLS)

## Author Contributions

**Conceptualization:** Yufang Li.

**Data curation:** Yanxia Guo.

**Formal analysis:** Yufang Li, Yanxia Guo, Du Chen.

**Investigation:** Yufang Li, Yanxia Guo.

**Methodology:** Yufang Li, Du Chen.

**Resources:** Yufang Li, Yanxia Guo.

**Software:** Du Chen.

**Supervision:** Du Chen.

**Validation:** Du Chen.

**Writing – original draft:** Yufang Li.

**Writing – review & editing:** Du Chen.

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
