## [Editor Report · Decision Letter 0]

2 Sep 2020

PONE-D-20-22710

Emergency mortality of medical patients was predicted by qSOFA score

PLOS ONE

Dear Dr. Chen,

Thank you for submitting your manuscript to PLOS ONE. After careful consideration, we feel that it has merit but does not fully meet PLOS ONE’s publication criteria as it currently stands. Therefore, we invite you to submit a revised version of the manuscript that addresses the points raised during the review process.

We look forward to receiving your revised manuscript.

Kind regards,

Marleen Smits, PhD

Academic Editor

PLOS ONE

Additional Editor Comments:

Dear Dr. Chen,

You have submitted a related manuscript to PLOS ONE entitled "Predictive value of qSOFA score for death in emergency department resuscitation room among adult trauma patients".

We have considered the overlap between the two papers in light of the journal's policy.

In your paper entitled “Emergency mortality of medical patients was predicted by qSOFA score” the predictive value of qSOFA for death in the ED was examined.

The second paper “Predictive value of qSOFA score for death in emergency department resuscitation room among adult trauma patients” presents a subgroup analysis of trauma patients in the ED resuscitation room. These patients were also included in the study described in the first paper.

Both analyses provide valuable information. However by splitting up the results into two separate papers, it seems that there are two studies that provide evidence for the predictive value of qSOFA, while in fact it is only one study. The study population of “Emergency mortality of medical patients was predicted by qSOFA score"" includes the study population of "Predictive value of qSOFA score for death in emergency department resuscitation room among adult trauma patients".

Therefore, we suggest you combine the two papers into one paper.

We look forward to receiving a major revision of “Emergency mortality of medical patients was predicted by qSOFA score", which includes the subgroup analysis of adult trauma patients in the ED resuscitation room.

Kind regards,

Marleen Smits
---

## [Author Response · Author response to Decision Letter 0]

28 Sep 2020

Dear Editor,

Thank you for reviewing the manuscript, we read your comments and suggestions carefully.

I'm sorry that the wording of the title of our manuscript may be inaccurate, resulting in your misunderstanding. 

In fact, Paper1 (Emergency mortality of medical patients was predicted by qSOFA score) and Paper2 (Predictive value of qSOFA score for death in emergency department resuscitation room among adult trauma patients) were two separate sample studies, conducted by two different researchers (YuFang Li, WenJuan Huang) respectively. The study samples of Paper1 were non-traumatic patients (n=11621) treated by physicians in the emergency room, while the study samples of Paper2 were trauma patients (n=1739) treated by surgeons in the emergency room. Therefore, the research sample of Paper2 is not a subgroup of the research sample of Paper 1, but two different research samples.

In addition, there was considerable discrepancy between their research methods. For example, only age and gender were included in the multivariate regression model for non-traumatic patients to calculate the adjusted OR in Paper 1, but RIS scores reflecting the severity of injuries were added as adjusted variables in Paper2. Therefore, the adjusted OR meanings of these two studies are different.

To sum up, the study population of Paper1 does not include the study population of Paper2, and the research methods are also not exactly the same, so the two papers cannot be combined.

In order to avoid misunderstanding, we decided to revise the title of Paper1 as Emergency mortality of internal medicine patients was predicted by qSOFA score.

Thank you again for all your hard work and generous help.

Kind regards,

Du Chen

---

## [Decision Letter · Decision Letter 1]

27 Oct 2020

PONE-D-20-22710R1

Emergency mortality of internal medicine patients was predicted by qSOFA score

PLOS ONE

Dear Dr. Chen,

Thank you for submitting your manuscript to PLOS ONE. After careful consideration, we feel that it has merit but does not fully meet PLOS ONE’s publication criteria as it currently stands. Therefore, we invite you to submit a revised version of the manuscript that addresses the points raised during the review process.

We look forward to receiving your revised manuscript.

Kind regards,

Marleen Smits, PhD

Academic Editor

PLOS ONE

Reviewers' comments:

Reviewer's Responses to Questions

**Comments to the Author**

1. If the authors have adequately addressed your comments raised in a previous round of review and you feel that this manuscript is now acceptable for publication, you may indicate that here to bypass the “Comments to the Author” section, enter your conflict of interest statement in the “Confidential to Editor” section, and submit your "Accept" recommendation.

Reviewer #1: (No Response)

Reviewer #2: (No Response)

2. Is the manuscript technically sound, and do the data support the conclusions?

Reviewer #1: No

Reviewer #2: Partly

3. Has the statistical analysis been performed appropriately and rigorously? 

Reviewer #1: Yes

Reviewer #2: Yes

4. Have the authors made all data underlying the findings in their manuscript fully available?

Reviewer #1: Yes

Reviewer #2: Yes

5. Is the manuscript presented in an intelligible fashion and written in standard English?

Reviewer #1: Yes

Reviewer #2: No

6. Review Comments to the Author

Reviewer #1: The study design and methods appear adequate to answer the research question. However the rationale of the research question and conclusions drawn from the results are inadequate in my opinion. The qSOFA was designed to assess patients with possible sepsis to quickly identify those with increased risk of adverse outcome. In this study the qSOFA is evaluated as a prediction tool for ED mortality in all non-traumatic patients. It is not explained what the clinical use can be. It is good practice in EDs to triage all patients upon arrival. The most important factor determining the triage level is the chance of (ED)mortality. It is not explained if the qSOFA is proposed as a triage system, or that it adds to the existing triage system. Furthermore, the qSOFA score is not compared to the NEWS (or other EWS), as the authors deem these to inconvenient to measure in the ED. However, the respiratory rate is the vital sign that is most time consuming to measure, and NEWS only adds point for the heart rate, SpO2 and body temperature, measurements that can be easily obtained during triage. The conclusion that qSOFA has good performance is relative. Most patients with a high score survive (598/672 patients with qSOFA ≥2), and most non-survivors presented with a qSOFA <2 (154/228 patients). To assess whether this is a good performance, the results should be compared to NEWS and/or triage categories. Lastly, the (revised) title is not clear: the term “emergency mortality” should best be avoided and “internal medicine patients” usually do not include patients presented with cardiac complaints. “Non-trauma patients” appears to be a better representation of the study population.

Reviewer #2: Chen et al. analyse the association between qSOFA score and early mortality in the emergency department in a large cohort of “internal medicine” patients.

I have some suggestions:

- English form is poor and needs revising by a native-english speaker.

- Abstract, Discussion: authors claim that their study is adequate “in predicting the emergency department (ED) mortality”. However, this is a retrospective cohort study, and authors can only observe an association between variables. In order to observe a prediction, a prospective or – better – a randomized study will be required.

- Abstract, Materials and Methods: “internal medicine patients” – please clarify: were these patients the ones admitted to an internal medicine department ? Or this term is adopted to identify generically “medical patients” ? Since authors adopt “ED mortality” as main outcome, I think that authors mean for “internal medicine patients” the ones with medical conditions, however this concept is ambiguous and not clear. Moreover, the analysed cohort is not well characterised, which is a major limitation for the interpretation of the results. Authors must define in the results which medical pathologies were considered (acute myocardial infarction ? sepsis ? septic shock ? acute heart failure ? atrial fibrillation ? pneumonia ? etc.) and which was the prevalence of outcome in each subgroup.

- Materials and Methods, Results: in particular, how many patients affected by infective diagnoses were included in the cohort ? how many with sepsis or septic shock ? was the outcome different in patients admitted for "infective" pathologies and "non-infective" pathologies ?

- Materials and Methods: the exclusion of ischemic stroke is a bias, in my opinion, since it is an acute medical condition burdened by a high mortality. However authors specified this as a limitation of the study. Were other neurologic conditions (status epilepticus, brain haemorrage, and so on) considered in this cohort ?

- Materials and Methods: another point is the absence of comorbidities: authors should at least consider the most common comorbidities in their analyses (COPD, chronic heart failure, cancer, atrial fibrillation, ischemic heart disease, etc.) since several studies associated the complexity of the patient to an increased risk of in-hospital death. This could radically modify the estimates performed in the logistic regression analysis.

- Materials and Methods: a Cox regression analysis considering the time-to-event (time-to-ED death) could considered instead of logistic regression analysis.

- Materials and Methods, Results: authors did not consider patients died after hospitalization from ED. I think, however, that they could present the datum of how many patients were admitted to an ICU, which could be a surrogate of their critical illness, as already done in other similar studies.

- Results, Materials and Methods, Discussion: this study is similar other published papers (for example: https://emj.bmj.com/content/35/6/350), in which qSOFA was observed to perform better than other scores in predicting ED mortality and critical illness. Authors should discuss on what is similar and what is different from what is already known and published.

7. PLOS authors have the option to publish the peer review history of their article (what does this mean?). If published, this will include your full peer review and any attached files.

Reviewer #1: No

Reviewer #2: No

---

## [Author Response · Author response to Decision Letter 1]

14 Jan 2021

To Reviewer #1

Dear reviewer，thank you very much for your review and suggestions.

Because the qSOFA score contains only three simple binary variables which are easy to get, we wanted to verify the predictive value of qSOFA score for the prognosis of non-trauma patients, including sepsis. The aim is to expand the scope of the qSOFA score and provide a simple assessment tool for non-trauma patients in the emergency department.

Your suggestion to compare the qSOFA score with the MEWS score has been adopted and revised in the manuscript. Our study shows that AUROC of qSOFA is lower than MEWS score, and the difference is statistically significant. But qSOFA scores have less content and are easier to calculate.The MEWS was miscalculated frequently, the probability as high as 18.2% (The 17th reference in the manuscript). Compared to MEWS, the AUC of qSOFA score is slightly lower, but still has good predictive value and can be used as a supplement to the existing triage system.

Finally, thank you for your suggestion to modify the title. We have modified the title as “Emergency mortality of non-trauma patients was predicted by the qSOFA score”.

To Reviewer #2

Dear reviewer，thank you very much for your review and suggestions.

--Abstract, Discussion: It is true, as you said, that although the qSOFA score is associated with ED mortality in non-trauma patients and shows a good prognostic performance, this study is just a retrospective study, a prospective will be required.

--Abstract, Materials and Methods: Your suggestion is very constructive, the concept of internal medical patients is really vague, we have modified the title as ”Emergency mortality of non-trauma patients was predicted by the qSOFA score”.Our aim is to expand the practicability of qSOFA score , especially in undifferentiated non-trauma patients, so we didn’t conduct a subgroup analysis based on medical pathologies.

--Abstract, Materials and Methods: The purpose of our study was to verify the predictive value for the prognosis of non-traumatic patients，expand the scope of qSOFA score and provide a simple assessment tool for all non-trauma patients in the emergency department. And studies have confirmed that the predictive validity of death in patients with and without suspected infection were similarly high (The 12th reference in the manuscript). So we did not deliberately distinguish between the infected and non-infected patients.

-- Materials and Methods: All the non-trauma patients included in this study were treated by physicians, while neurological diseases were treated by neurologist, so they were not included in the study.

-- Materials and Methods: This is a retrospective study that extracted information from the hospital emergency electronic record system. The time of onset, complications, ICU admission, and other information were not recorded. To sum up, this study does have some limitations in design.

-- Materials and Methods: The Cox regression you proposed instead of logistic regression was indeed very good. However, since our system only has the time of register, we cannot accurately calculate the time from onset to death. After careful consideration, logistic regression analysis was still used.

-- Materials and Methods, Results: The only outcome of our study was ED mortality, so we did not consider post-admission deaths and ICU admissions of non-traumatic patients. 

-- Results, Materials and Methods, Discussion: In the discussion section, we have compared the results with those of some published studies, especially those that, like our studies, target people who are not sepsis(References to articles 6-8 and 12-13 in the manuscript). The predictive performance of qSOFA score and MEWS score was also compared in the results and discussion section.

---

## [Decision Letter · Decision Letter 2]

28 Jan 2021

PONE-D-20-22710R2

Emergency mortality of non-trauma patients was predicted by the qSOFA score

PLOS ONE

Dear Dr. Chen,

Thank you for submitting your manuscript to PLOS ONE. After careful consideration, we feel that it has merit but does not fully meet PLOS ONE’s publication criteria as it currently stands. Therefore, we invite you to submit a revised version of the manuscript that addresses the points raised during the review process.

Please address the comment of reviewer 1 that the MEWS gives a better prediction, which should lead to an adjustment of the conclusions of the manuscript. Moreover, please have a native English speaker check the language in the paper.

We look forward to receiving your revised manuscript.

Kind regards,

Marleen Smits, PhD

Academic Editor

PLOS ONE

Reviewers' comments:

Reviewer's Responses to Questions

**Comments to the Author**

1. If the authors have adequately addressed your comments raised in a previous round of review and you feel that this manuscript is now acceptable for publication, you may indicate that here to bypass the “Comments to the Author” section, enter your conflict of interest statement in the “Confidential to Editor” section, and submit your "Accept" recommendation.

Reviewer #1: (No Response)

Reviewer #2: All comments have been addressed

2. Is the manuscript technically sound, and do the data support the conclusions?

Reviewer #1: No

Reviewer #2: Partly

3. Has the statistical analysis been performed appropriately and rigorously? 

Reviewer #1: Yes

Reviewer #2: Yes

4. Have the authors made all data underlying the findings in their manuscript fully available?

Reviewer #1: Yes

Reviewer #2: Yes

5. Is the manuscript presented in an intelligible fashion and written in standard English?

Reviewer #1: Yes

Reviewer #2: No

6. Review Comments to the Author

Reviewer #1: I acknowledge the additional analysis of the MEWS by the authors. However, in accordance with other research, the MEWS gives a better prediction, which is in my opinion reason to adjust the conclusions of the manuscript. The AUROC is not the most indicative measure to assess the added value. The sens, spec, PPV and NPV at the cutoff point used in clinical practice are more important. For qSOFA a score of 2 or higher is used to identify high risk patients. At this cutoff point sensitivity is 32% and specificity 96%. According to the ROC curve, at the optimal cutoff point of MEWS sens is about 50% with spec of 97-98%. This is a major difference, not shown in the relatively small difference in AUROC. The arguments that MEWS is to complicated to use in the ED and scores are often miscalculated are not valid when the score are automatically calculated which is becoming more and more standard practice. The manuscript school be revised, either more in support of the use of MEWS/NEWS (early warning scoring systems can vary substantially and may also be compared), or as a validation of the prediction of mortality of qSOFA (without concluding it should be used in triage).

Reviewer #2: I have read the authors' responses and the reviewed article. Authors answered to most of the questions and the overall quality of the paper improved after revision. However, English form is still not fair for publication in an international journal. I recommend to deeply revise the English form (best if the revision is done by a native-English-speaker) in all the sections of the paper.

7. PLOS authors have the option to publish the peer review history of their article (what does this mean?). If published, this will include your full peer review and any attached files.

Reviewer #1: No

Reviewer #2: No

---

## [Author Response · Author response to Decision Letter 2]

7 Feb 2021

To Reviewer #1

Dear reviewer，thank you very much for your review and suggestions.

We have calculated best cutoff values, sensitivity and specificity for qSOFA score and MEWS score based on the Yoden index (listed in the results section of the manuscript). 

Indeed, due to the widespread use of computers, the complexity and error-prone nature of MEWS score are negligible problems, so we clearly suggest that MEWS was a better choice as a triage tool.

According to the results of the study,the qSOFA score can be used as a tool to predict ED mortality in non-trauma patients.

To Reviewer #2

Dear reviewer，thank you very much for your review and suggestions.

The English form has been deeply modified.

---

## [Editor Report · Decision Letter 3]

10 Feb 2021

Emergency mortality of non-trauma patients was predicted by the qSOFA score

PONE-D-20-22710R3

Dear Dr. Chen,

We’re pleased to inform you that your manuscript has been judged scientifically suitable for publication and will be formally accepted for publication once it meets all outstanding technical requirements.

Kind regards,

Marleen Smits, PhD

Academic Editor

PLOS ONE
---

## [Editor Report · Acceptance letter]

15 Feb 2021

PONE-D-20-22710R3 

Emergency mortality of non-trauma patients was predicted by qSOFA score 

Dear Dr. Chen:

I'm pleased to inform you that your manuscript has been deemed suitable for publication in PLOS ONE. Congratulations! Your manuscript is now with our production department. 

Kind regards, 

on behalf of

Dr. Marleen Smits 

Academic Editor

PLOS ONE